# Correcting in Hindsight: Editing Past Key-Value States for Robust LLM Reasoning

**Mengfei Zhang** [* 1]  **Yu Mi** [* 1]  **Leijing Zhou** [1]

## Abstract

Autoregressive Large Language Models (LLMs) often fail in complex reasoning because early-stage errors remain uncorrectable in subsequent steps—a limitation fundamentally rooted in the inherent irreversibility of the Transformer architecture. In this paper, we propose HEdit, a lightweight reasoning enhancement paradigm that equips models with a "hindsight-like" capability for dynamic error correction during generation. Our core insight involves deconstructing reasoning failures into two pivotal stages: latent representational biases emerging at logical anchors, and the subsequent eruption of explicit cognitive dissonance at trigger points. Based on these observations, the HEdit framework detects internal inconsistency signals at trigger points in real-time, actively backtracks to critical anchors, and utilizes a lightweight trainable editor to precisely refine their Key-Value (KV) caches. This mechanism effectively breaks the unidirectional constraints of autoregressive inference. Empirical results demonstrate that HEdit significantly enhances the performance of various models on mathematical reasoning tasks—with average accuracy improvements ranging from 2.2% to 10.8%—while maintaining extremely low overhead (add parameters $< 0.5\%$). HEdit provides a dynamic, pluggable and lightweight solution, making it particularly beneficial for users in low-resource environments. Our code can be found at github: https://github.com/Zmfei/hedit

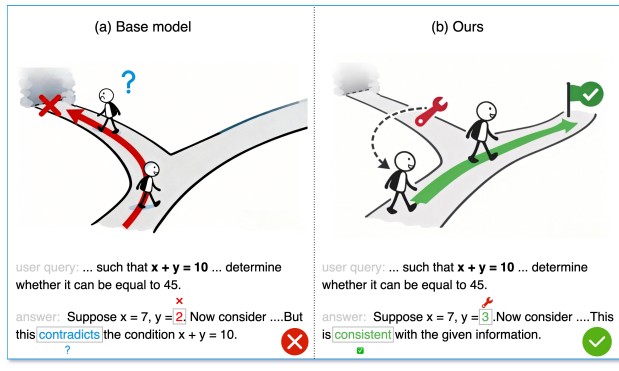

*Figure 1.* **Autoregressive reasoning failures and our solution.** (a) In standard autoregressive decoding, an early error at a critical anchor token (red) is silently propagated until it manifests as uncertainty or contradiction at a later trigger point (blue), where the model cannot revise its past reasoning. (b) Our method detects such trigger points, locates the responsible anchor token, retroactively edits its key-value states, and resumes reasoning along a corrected trajectory.

## 1. Introduction

Large Language Models (LLMs) have made remarkable breakthroughs in complex tasks such as mathematics and logic through the introduction of explicit reasoning processes, notably Chain-of-Thought (CoT) (Wei et al., 2022). Large Reasoning Models (LRMs), represented by OpenAI's o1 (Jaech et al., 2024) and DeepSeek-R1 (Guo et al., 2025), have further demonstrated that scaling test-time compute can significantly bolster the accuracy of final answers.

To bolster the reasoning capabilities of LLMs, existing research generally follows two trajectories. The first involves training-free methods, such as CoT prompting or Self-Consistency (Wang et al., 2022), which aim to elicit the model's intrinsic logical capabilities. The second encompasses training-driven strategies, including Supervised Fine-Tuning (SFT), RL, or Recurrent Transformers, which improve reasoning capabilities by optimizing parameters or increasing computational depth. Despite their success, we argue that these methods remain constrained by a fundamental flaw in the Transformer architecture: **Autoregressive**

---

[*]Equal contribution  [1]School of Software Technology, Zhejiang University. Correspondence to: Leijing Zhou <leijing@zju.edu.cn>.

*Proceedings of the 43rd International Conference on Machine Learning*, Seoul, South Korea. PMLR 306, 2026. Copyright 2026 by the author(s).

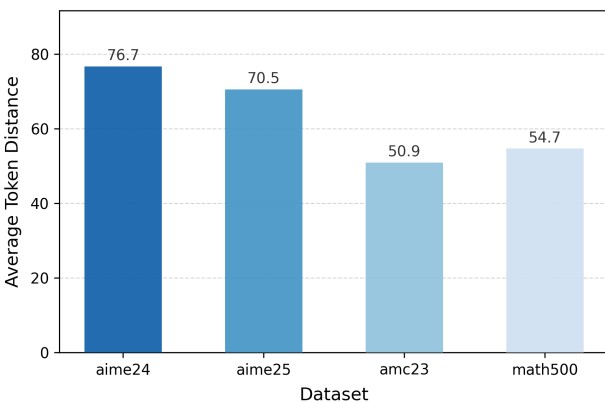

*Figure 2.* **Average Token Distance between Anchors and Triggers.** Based on a manual audit of Qwen3-8B reasoning failures, the chart illustrates the mean token distance from the anchor (the error inception) to the failure Trigger across various datasets.

**Irreversibility**. Once a token is generated, its corresponding Key/Value (KV) states are committed to the cache and utilized as "fixed memories" in subsequent inference. Even if the model realizes a prior misunderstanding in later steps, these early internal states remain immutable.

This irreversibility gives rise to a distinctive and often overlooked failure mode in autoregressive reasoning, which can be decomposed into two sequential phases: *error latency* and *reasoning collapse*. **In the first phase (error latency)**, subtle representational biases introduced at critical logical junctions cause the reasoning trajectory to deviate from the correct path. Importantly, these early errors are typically insidious: the model continues to produce multiple seemingly plausible intermediate steps, masking the underlying inconsistency.**In the second phase (reasoning collapse)**, the accumulated bias is eventually exposed when a later reasoning step depends on the earlier corrupted information. At this trigger point, the model can no longer maintain internal consistency, leading to abrupt uncertainty, contradiction, and a cascading breakdown of the reasoning chain. Figure 1 illustrates this delayed failure pattern in standard autoregressive decoding and contrasts it with our hindsight-based correction mechanism.

For this "compromised foundation" phenomenon, existing methods—struggle to fundamentally reverse the situation, because they implicitly assume the current context is valid, failing to rectify the KV cache already "contaminated" by early-stage errors.

In this paper, by systematically deconstructing the aforementioned failure process, we present three key insights into the model's internal behavior:

- **Logical Anchors:** In a reasoning chain, not all to-

kens carry equal weight. Only a few critical tokens—representing numerical values, constraints, or pivotal logic—serve as anchors. Once their internal representations are biased, the entire reasoning path shifts systematically.

- **Failure Triggers:** At the moment of reasoning collapse, models often enter a state of "cognitive dissonance": characterized by violent oscillations in hidden layer representations across adjacent layers and high uncertainty in output distributions. This suggests the model is not "confidently wrong" but is aware of a reasoning anomaly yet lacks a mechanism for correction. We define the token position where this occurs as the trigger.

- **Latency in Error Discovery:** Crucially, the trigger is not where the error originates, but rather a delayed manifestation of a latent error. The root cause typically resides in an earlier anchor. This latency is quantified in Figure 2, confirming a persistent token distance between the anchor and the subsequent trigger.

Based on these insights, we propose Hindsight Editing (HEdit), a lightweight dynamic reasoning enhancement paradigm. HEdit equips models with human-like "hindsight": when a model senses a reasoning dilemma at a trigger, it actively backtracks and applies minimal corrections to the internal representations of anchors. Specifically, our approach consists of: (1) a training-free anchor identification mechanism; (2) a training-free trigger detection mechanism; and (3) a small, trainable editor (MLP) that refines the KV cache of anchors using the current state. Figure 3 provides an overview of the Sense-Backtrack-Edit-Regenerate workflow, illustrating how these components interact to retroactively correct latent reasoning errors during inference.

HEdit offers several distinct advantages: it does not modify the original model parameters and intervenes in only a tiny fraction of internal states only when necessary (on average, only 28.3% of samples require intervention), ensuring negligible computational overhead while preserving general capabilities. Experiments across multiple mathematical benchmarks, including AIME24, AIME25, AMC23, and MATH500(Balunović et al., 2025; Lightman et al., 2023), demonstrate that HEdit significantly improves the reasoning accuracy of models across various scales and architectures (Qwen, Llama, 0.6b to 14b)(Yang et al., 2025; Dubey et al., 2024; Guo et al., 2025), with an average accuracy boost of 2.2%–10.8%, consistently outperforming baselines like CoT and direct re-reasoning at anchors.

Our primary contributions are summarized as follows:

- We systematically characterize a critical failure mechanism in LRMs, highlighting that reasoning collapse

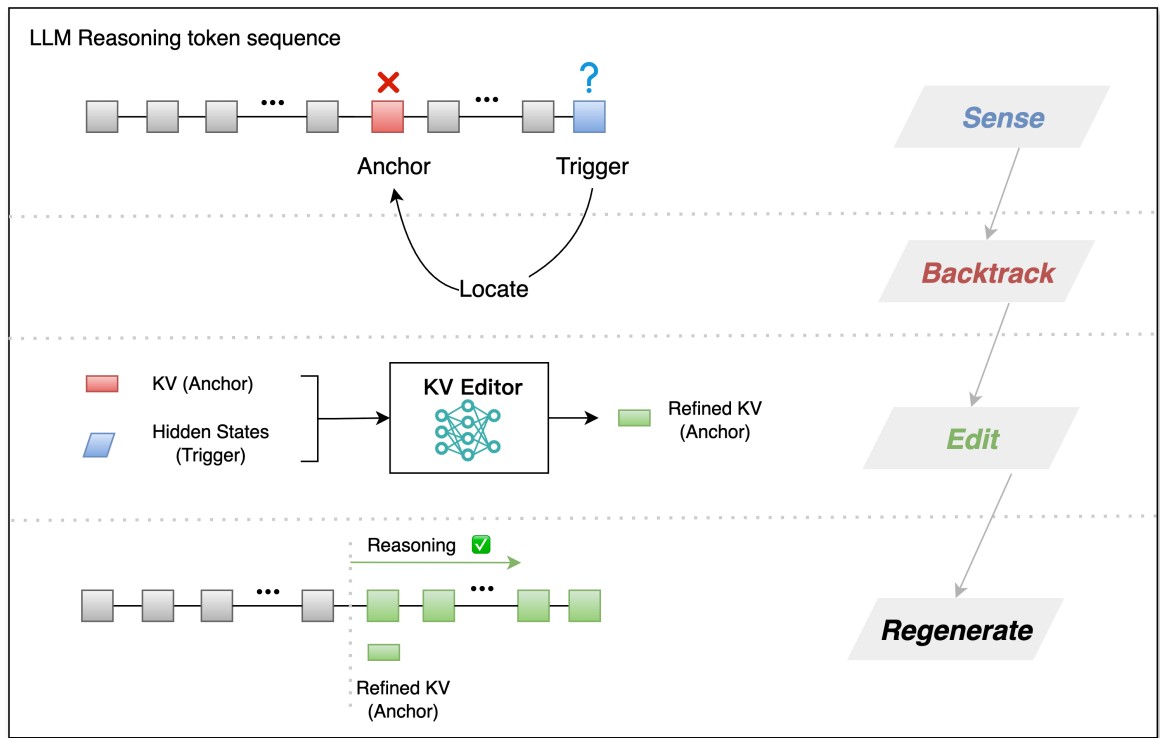

*Figure 3.* Overview of the **HEdit** Framework. HEdit **senses** reasoning failures by monitoring diagnostic signals at both logical Anchors and the failure Trigger. Upon detecting a trigger, it **backtracks** to the most recent Anchor, **edits** the corrupted KV cache using hindsight information from the Trigger, and **regenerates** the sequence to steer the reasoning trajectory toward a correct conclusion.

stems from representational biases at key anchors and revealing the latency effect between error inception and manifestation.

- We propose HEdit, a new reasoning paradigm that implements a "Sense-Backtrack-Edit-Regenerate" loop, effectively overcoming the inherent constraints of autoregressive irreversibility.

- Without requiring parameter fine-tuning, HEdit demonstrates robust generalizability and significant performance gains across diverse architectures, parameter scales, and mathematical reasoning tasks.

**Conflict of Interest Disclosure** The authors declare that there are no conflicts of interest to disclose.

## 2. Related Work

### 2.1. Reasoning via Prompting and Search

Chain-of-Thought (CoT) prompting (Wei et al., 2022) elicits reasoning through explicit intermediate steps. This paradigm is further extended by multi-path sampling and search algorithms, such as Self-Consistency (Wang et al.,

2022), Tree-of-Thoughts (Yao et al., 2023), and Graph-of-Thoughts (Besta et al., 2024). These methods expand the exploration space and enhance reasoning stability by aggregating results from multiple trajectories or performing tree/graph-based searches at the token level.

### 2.2. Learning-based Reasoning Optimization

A separate line of research focuses on bolstering reasoning through Supervised Fine-Tuning (SFT) and Reinforcement Learning (RL). Instruction-tuning efforts (Wei et al., 2021; Ouyang et al., 2022; Muennighoff et al., 2023; Taori et al., 2023) align models with high-quality reasoning data. Meanwhile, RL algorithms such as PPO (Schulman et al., 2017) and GRPO (Shao et al., 2024) optimize policy correctness. Recent models like DeepSeek-R1 (Guo et al., 2025) and Kimi k1.5 (Team et al., 2025) leverage large-scale RL to induce sophisticated logical deduction and extended CoT by scaling test-time computation.

### 2.3. Adaptive and Recurrent Computation

To increase computational depth per token, recent studies explore structural modifications to the Transformer. Looped

Transformers (Saunshi et al., 2025; Geiping et al., 2025) and Ponder (Zeng et al., 2025) introduce latent iterations or recurrent computations to refine internal representations. Similarly, Think-at-Hard (TaH) (Fu et al., 2025) dynamically allocates additional computation to difficult decision points during the forward pass to enhance the depth of "thinking".

**The HEdit Distinction.** Unlike the aforementioned methods that focus on expanding search paths, scaling parameters, or increasing per-token depth, HEdit addresses a critical structural bottleneck: the irreversibility of early erroneous context in autoregressive inference. Current reasoning frameworks are inherently forward-only and lack a mechanism to rectify historical states once a logic error is committed to the context. HEdit is the first to explicitly model and rectify Key-Value (KV) representations of pivotal tokens. By sensing reasoning dilemmas and backtracking to update "latent misunderstandings", HEdit enables the model to retroactively recover from errors and restore the reasoning trajectory.

## 3. Methodology

### 3.1. Problem Formalization and Overview

Given an autoregressive language model $M$ and an input sequence $x$, the model generates a reasoning chain token-by-token. At each step $t$, the model produces the hidden representation $h_t$ of current token. Due to the unidirectional constraint of the Transformer architecture, once an erroneous Key-Value (KV) state is generated at an early position $i < t$, this error is solidified in the cache and propagates through subsequent layers and steps, inevitably leading to reasoning failure.

We propose the HEdit (Hindsight Editing) framework to break this irreversibility. The framework consists of three synergistic modules:

- **Anchor Detection:** Identifies "pivotal" tokens that carry significant logical weight for future reasoning.

- **Trigger Monitoring:** Real-time sensing of internal "cognitive dissonance" signaling reasoning inconsistency.

- **Retroactive Editor:** A lightweight module that refines the KV cache of historical anchors using the current state when a trigger is activated.

### 3.2. Reasoning Diagnostics: Anchors and Triggers

#### 3.2.1. DEFINING LOGICAL ANCHORS

Logical anchors are the pivotal tokens that encode core logic and are discriminatively invoked by subsequent steps. To identify these nodes, we evaluate tokens across two dimensions: structural selectivity and semantic density.

**1. Attention Selectivity (Structural Dimension):**
Rather than using raw attention frequency—which is often biased by the "Attention Sink" effect (Xiao et al., 2023) where initial tokens attract spurious attention—we measure the Attention Score Variance to identify tokens that are utilized for specific logical dependencies:

$$\text{Score}_{attn}(i) = \text{Var}(\{\alpha_{t,i}\}_{t=i+1}^{T}) \tag{1}$$

where $\alpha_{t,i}$ is the attention score assigned to position $i$ at step $t$. A high variance indicates that a token is not "blindly" prioritized but is sharply attended to only when specific logical needs arise.

**2. Information Update Ratio (Semantic Dimension):**
To filter out non-semantic functional tokens (e.g., punctuation) that may still exhibit high attention variance, we quantify the Information Density of each token via its relative representational shift within the Feed-Forward Network (FFN) layers:

$$\text{Score}_{ffn}(i) = \frac{|\text{FFN}(\mathbf{h}_i) - \mathbf{h}_i|}{|\mathbf{h}_i|} \tag{2}$$

Intuitively, semantically rich tokens undergo significant transformations in the FFN space, whereas non-semantic functional tokens remain relatively static.

**Definition:** By synthesizing these structural and semantic constraints, we formally define the set of anchor tokens $\mathcal{A}$ as the intersection of the Top-$k$ candidates from both metrics:

$$\mathcal{A} = \text{Top-}k(\text{Score}_{attn}) \cap \text{Top-}k(\text{Score}_{ffn}) \tag{3}$$

#### 3.2.2. DEFINING FAILURE TRIGGERS

Trigger Points refer to the specific moments during the generation process when the model perceives a breach of logical self-consistency. To precisely localize these moments and filter out non-logical fluctuations, we identify triggers by synthesizing the following two metrics:

**1. Prediction Uncertainty:** This is quantified by the Shannon entropy of the model's output distribution at the $t$-th token:

$$H_t = -\sum_{v \in \mathcal{V}} p_t(v) \log p_t(v) \tag{4}$$

where $p_t$ denotes the output probability distribution at step $t$, and $\mathcal{V}$ is the vocabulary. A higher $H_t$ value indicates that the model is confounded by the current continuation.

**2. Cognitive Dissonance:** Relying solely on predictive entropy introduces significant noise; for instance, punctuation or syntactic tokens inherently possess high entropy due to various plausible alternatives, which does not necessarily signify a logical conflict. To isolate genuine logical contradictions, we calculate the negative cosine similarity between

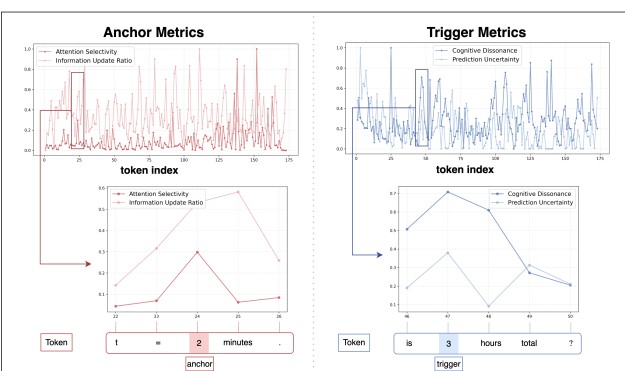

*Figure 4.* **Case Study of Anchor and Trigger Identification.** The line charts track the real-time variation of diagnostic metrics for a Qwen3-8B reasoning sequence. Magnified insets provide a detailed view of the specific tokens identified as Anchors (left) and Triggers (right).

hidden states across layers:

$$- \text{Sim}(t) = - \cos\left(\mathbf{h}_t^l, \mathbf{h}_t^{l-1}\right) \tag{5}$$

where $\mathbf{h}_t^l$ represents the hidden state at the $l$-th layer for the $t$-th token. Empirical observations suggest that for syntactic or functional tokens, cross-layer representations remain stable (yielding high similarity). Conversely, when the model encounters logical contradictions, its representations fluctuate significantly across layers (resulting in a sharp drop in similarity), reflecting internal "cognitive dissonance".

**Definition:** By synthesizing these metrics, we formally define the set of trigger points $\mathcal{T}$ as those ranking in the Top-$k$ for both negative cross-layer similarity and prediction entropy:

$$\mathcal{T} = \text{Top-}k(-\text{Sim}) \cap \text{Top-}k(H) \tag{6}$$

The real-time variation of these metrics is visualized in Figure 4.

### 3.3. Retroactive KV Editor

The core of HEdit is a lightweight MLP editor $\mathcal{M}$, which is extremely parameter-efficient (comprising $< 0.5\%$ of the backbone model's parameters) and consists of only two hidden layers.

- **Input:** The hidden state $\mathbf{h}_\mathcal{T}$ at the trigger point $\mathcal{T}$ and the original cache $\text{KV}_\mathcal{A}$ at the anchor point $\mathcal{A}$.

- **Output:** The rectified anchor cache $\text{KV}_{new}$.

**Self-Supervised Training Trajectory:**
To endow the editor with error-correction capabilities without requiring external human annotations, we employ a self-supervised training paradigm:

**1. Trajectory Construction:** We randomly sample 1k problems from GSM8K and perform 10 decoding runs per sample. Only the reasoning paths that lead to the correct final answer are retained as gold trajectories.

**2. Counterfactual Perturbation:** At the identified anchor $\mathcal{A}$, we forcibly inject an error by randomly sampling a token from the Top-2 to Top-5 predicted probabilities instead of the greedy Top-1. This counterfactual intervention deliberately pushes the model onto an erroneous reasoning track.

**3. Sample Collection:** We continue generation along the erroneous path until completion. We retain only those samples that result in an incorrect answer, identify their trigger points $\mathcal{T}$, and record the triplets $(\mathbf{h}_\mathcal{T}, \text{KV}_\mathcal{A}^{\text{bad}}, \text{KV}_\mathcal{A}^{\text{gold}})$, where $\text{KV}_\mathcal{A}^{\text{bad}}$ and $\text{KV}_\mathcal{A}^{\text{gold}}$ represent the anchor caches for the erroneous and gold paths, respectively.

**4. Robustness via Identity Mapping:** To prevent the editor from being over-sensitive to benign fluctuations, we incorporate correct reasoning paths into the training set. The editor is required to perform an identity mapping for these samples.

**Loss Function** The training objective is to minimize the Mean Squared Error (MSE) between the rectified KV cache and the gold KV cache:

$$\mathcal{L}_{\text{edit}} = \left\| \mathcal{M}(\mathbf{h}_\mathcal{T}, \text{KV}_\mathcal{A}^{\text{bad}}) - \text{KV}_\mathcal{A}^{\text{gold}} \right\|_2^2 \tag{7}$$

As illustrated in figure 5, on the test set's erroneous paths, the trained editor effectively shifts the contaminated $\text{KV}_\mathcal{A}^{\text{bad}}$ toward the direction of $\text{KV}_\mathcal{A}^{\text{gold}}$. This demonstrates its generalization capability in performing real-time reasoning rectification.

### 3.4. The HEdit Inference Framework

The HEdit execution flow follows a Sense-Backtrack-Edit-Regenerate cycle:

**1. Sense:** Generate tokens normally while dynamically updating anchor and trigger scores for all historical positions.

**2. Backtrack:** Once a trigger signal is detected at step $T$, the system identifies the most recent preceding anchor $i \in \mathcal{A}$.

**3. Edit:** The editor $\mathcal{M}$ is invoked to refine the anchor's KV cache: $KV_i \leftarrow \widehat{KV}_i$.

**4. Regenerate:** Predictions following the trigger are discarded. The model resumes generation from the anchor position $i$ using the rectified memory, effectively restoring the reasoning trajectory.

The full framework is shown in Figure 3.

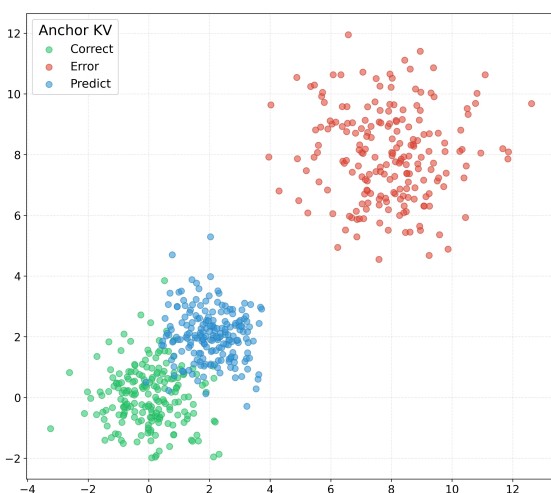

*Figure 5.* **t-SNE Visualization of KV Cache Rectification.** This plot illustrates the distribution of anchor KV caches in the latent space for Qwen3-8B. It demonstrates the ability of the editor to shift "bad" anchor caches ($\text{KV}_{\mathcal{A}}^{\text{bad}}$) significantly toward their corresponding gold counterparts ($\text{KV}_{\mathcal{A}}^{\text{gold}}$) on the test set, effectively bridging the representational gap caused by reasoning errors.

# 4. Experiments

We conduct rigorous experiments to evaluate the effectiveness and superiority of the HEdit paradigm. This section is designed to systematically address the following four Research Questions (RQs):

- RQ1 (Effectiveness): Can HEdit consistently improve the mathematical reasoning capabilities of models across different architectures and scales?

- RQ2 (Necessity of Mechanism): What is the advantage of dynamic KV cache correction compared to simple backtracking and re-reasoning?

- RQ3 (Ablation Study): How do the individual modules (Anchor/Trigger detection, Editor) contribute to the final performance?

- RQ4 (Efficiency): Does HEdit meet the lightweight design goals in terms of intervention frequency and inference latency?

## 4.1. Experimental Setup

Models: To evaluate generalizability, we test HEdit across several mainstream model families and scales:

- Qwen3 Series: 0.6B, 1.7B, 4B, 8B, and 14B.

- Llama Series: Llama-3.1-8B-Instruct.

- Reasoning-Enhanced Models: DeepSeek-R1-Distill-Qwen/Llama (ranging from 1.5B to 14B).

Datasets: We select competitive mathematical benchmarks requiring multi-step reasoning: AIME 2024, AIME 2025, AMC23, and MATH500.
Baselines:

- Base Model: Standard autoregressive generation.

- Chain-of-Thought (CoT): Prompting the model to generate intermediate reasoning steps.

- Naive Backtracking: Upon detecting a trigger, the model backtracks to the anchor and re-reasoning without modifying the KV cache.

Metrics and Implementation: We report Pass@k accuracy. The maximum generation length is set to 16,384 tokens. For each problem, we sample 4 to 16 outputs with a sampling temperature of 0.7. Details of the experimental setup are provided in Appendix A.

## 4.2. Main Results: Effectiveness Across Tasks (RQ1)

Table 1 summarizes the accuracy of HEdit across various models and datasets (See Appendix B for complete results). Our primary observations are as follows:
**1. Robust Performance Gains:** HEdit achieves consistent and significant performance improvements across all settings, with average gains ranging from 2.2% to 10.8%. This validates HEdit as a highly effective plug-and-play intervention mechanism.
**2. Scaling Trends of Reasoning Enhancements:** We observe that smaller models (e.g., 0.6B/1.7B) tend to receive larger relative boosts. A plausible explanation is that smaller models are more susceptible to error accumulation in long-chain reasoning, thus benefiting more from an external "hindsight" correction mechanism.
**3. Synergy with RL-tuned Models:** Notably, HEdit provides additional gains even for the DeepSeek-R1-Distill series, which already possesses strong reasoning capabilities via Reinforcement Learning. This suggests that the "Error Latency and Collapse" problem we address is a distinct challenge from basic reasoning ability, and HEdit serves as a complementary runtime safeguard.

## 4.3. Mechanism Analysis: HEdit vs. Naive Backtracking (RQ2)

Table 1 also compares HEdit with Naive Backtracking. A key finding is that simply backtracking and re-reasoning yields marginal improvements (averaging only +0.6% to +1.9%).
This result strongly supports the core design of HEdit. Naive

*Table 1.* Main Result: Performance comparison on various datasets.

| Model | Method | MATH500 | AIME24 | AIME25 | AMC23 | Average |
|---|---|---|---|---|---|---|
| Qwen3-0.6B | Original | 43.2 | 4.3 | 4.6 | 17.9 | 17.5 |
| Qwen3-0.6B | + CoT | 45.5(+2.3) | 7.1(+2.8) | 6.0(+1.4) | 19.2(+1.3) | 19.4(+1.9) |
| Qwen3-0.6B | + Naive BT | 43.6(+0.4) | 5.7(+1.4) | 4.9(+0.3) | 18.4(+0.5) | 18.1(+0.6) |
| Qwen3-0.6B | **+ Ours** | **56.1(+12.9)** | **11.1(+6.8)** | **16.5(+11.9)** | **29.3(+11.4)** | **28.2(+10.8)** |
| Qwen3-8B | Original | 73.7 | 65.6 | 61.3 | 65.3 | 66.5 |
| Qwen3-8B | + CoT | 74.8(+1.1) | 70.5(+4.9) | 64.7(+3.4) | 69.8(+4.5) | 70.0(+3.5) |
| Qwen3-8B | + Naive BT | 74.1(+0.4) | 67.4(+1.8) | 61.6(+0.3) | 67.7(+2.4) | 67.7(+1.2) |
| Qwen3-8B | **+ Ours** | **80.8(+7.1)** | **72.1(+6.5)** | **65.1(+3.8)** | **73.3(+8.0)** | **72.8(+6.3)** |
| Qwen3-14B | Original | 86.6 | 68.4 | 66.7 | 70.2 | 73.0 |
| Qwen3-14B | + CoT | 91.5(+4.9) | 69.6(+1.2) | 70.9(+4.2) | 74.4(+4.2) | 76.6(+3.6) |
| Qwen3-14B | + Naive BT | 89.3(+2.7) | 70.4(+2.0) | 67.2(+0.5) | 72.5(+2.3) | 74.8(+1.9) |
| Qwen3-14B | **+ Ours** | **93.0(+6.4)** | **73.7(+5.3)** | **73.0(+6.3)** | **75.5(+5.3)** | **78.8(+5.8)** |
| Llama3.1-8B-Instruct | Original | 47.6 | 5.1 | 1.9 | 8.6 | 15.8 |
| Llama3.1-8B-Instruct | + CoT | 49.1(+1.5) | 7.1(+2.0) | 2.7(+0.8) | 11.3(+2.7) | 17.6(+1.8) |
| Llama3.1-8B-Instruct | + Naive BT | 48.8(+1.2) | 7.0(+1.9) | 2.0(+0.1) | 10.9(+2.3) | 17.2(+1.4) |
| Llama3.1-8B-Instruct | **+ Ours** | **56.5(+8.9)** | **7.9(+2.8)** | **3.9(+2.0)** | **17.0(+8.4)** | **21.3(+5.5)** |
| DeepSeek-R1-Distill-Qwen-1.5B | Original | 83.2 | 27.5 | 25.4 | 55.4 | 47.9 |
| DeepSeek-R1-Distill-Qwen-1.5B | + CoT | 85.0(+1.8) | 28.8(+1.3) | 28.1(+2.7) | 58.5(+3.1) | 50.1(+2.2) |
| DeepSeek-R1-Distill-Qwen-1.5B | + Naive BT | 84.2(+1.0) | 29.8(+2.3) | 26.6(+1.2) | 58.0(+2.6) | 49.6(+1.8) |
| DeepSeek-R1-Distill-Qwen-1.5B | **+ Ours** | **91.2(+8.0)** | **30.9(+3.4)** | **36.9(+11.5)** | **61.8(+6.4)** | **55.2(+7.3)** |
| DeepSeek-R1-Distill-Qwen-14B | Original | 93.1 | 62.7 | 39.5 | 85.2 | 70.1 |
| DeepSeek-R1-Distill-Qwen-14B | + CoT | 94.0(+0.9) | 64.7(+2.0) | **44.2(+4.7)** | 88.7(+3.5) | **72.8(+2.6)** |
| DeepSeek-R1-Distill-Qwen-14B | + Naive BT | 93.7(+0.6) | 63.8(+1.1) | 40.3(+0.8) | 86.2(+1.0) | 71.0(+0.9) |
| DeepSeek-R1-Distill-Qwen-14B | **+ Ours** | **94.3(+1.2)** | **65.1(+2.4)** | 41.1(+1.6) | **88.8(+3.6)** | 72.3(+2.2) |
| DeepSeek-R1-Distill-Llama-8B | Original | 89.7 | 44.8 | 30.8 | 79.2 | 61.1 |
| DeepSeek-R1-Distill-Llama-8B | + CoT | 91.3(+1.6) | 47.3(+2.5) | **35.7(+4.9)** | 83.9(+4.7) | 64.6(+3.4) |
| DeepSeek-R1-Distill-Llama-8B | + Naive BT | 91.1(+1.4) | 45.7(+0.9) | 32.0(+1.2) | 82.1(+2.9) | 62.7(+1.6) |
| DeepSeek-R1-Distill-Llama-8B | **+ Ours** | **91.4(+1.7)** | **47.6(+2.8)** | 35.6(+4.8) | **87.7(+8.5)** | **65.6(+4.5)** |

re-computation often replicates the initial conditions that led to the error, causing the model to likely fall back into a similar erroneous trajectory. In contrast, HEdit integrates the "distress" signal from the trigger point to perform a targeted correction of historical KV caches. This effectively creates a new, corrected starting point for reasoning, facilitating a backward flow of information that transforms late-stage failure awareness into early-stage decision refinement.

### 4.4. Ablation Study: Component Contributions (RQ3)

We conduct an ablation study by systematically replacing key components (results in Table 2, See Appendix C for complete results):

- **Trigger Detection:** Replacing our entropy-and-similarity-based diagnostic with a random trigger leads to a performance drop of 5.5%–6.6%. This underscores the importance of precisely locating "cognitive dissonance" to ensure timely intervention.

- **Anchor Localization:** When anchors are replaced with random token positions, the performance gain is substantially diminished (dropping by 5.9%–11.9%).

This confirms that corrections must target the pivotal nodes carrying critical logical weight.

- **Editor Functionality:** Replacing the trained MLP editor with a random perturbation module results in severe performance degradation (dropping by 5.9%–25%). This proves that the gains stem from the editor's learned mapping from "current confusion" to "historical bias correction" rather than arbitrary state modification.

### 4.5. Efficiency and Overhead Analysis (RQ4)

We further analyze the practical overhead of HEdit during deployment:

- **Intervention Frequency:** As shown in Table 3, the average intervention rate for HEdit on Qwen models is 28.3% across datasets. This means over 70% of samples require no additional operations, highlighting its "on-demand" efficiency.

- **Inference Latency:** Thanks to the extremely small parameter count of the editor ($< 0.5\%$ of the back-

*Table 2.* Ablation study results.

| Model | Method | MATH500 | AIME24 | AIME25 | AMC23 | Average |
|---|---|---|---|---|---|---|
| Qwen3-0.6B | **Ours** | **56.1** | **11.1** | **16.5** | **29.3** | **28.2** |
| Qwen3-0.6B | + Random Trigger | 49.7(-6.4) | 5.6(-5.5) | 10.6(-5.9) | 22.8(-6.5) | 22.2(-6.0) |
| Qwen3-0.6B | + Random Anchor | 46.1(-10.0) | 2.8(-8.3) | 5.2(-11.3) | 18.8(-10.5) | 18.2(-10.0) |
| Qwen3-0.6B | + Random KV Editor | 40.0(-16.1) | 0.0(-11.1) | 0.0(-16.5) | 11.6(-17.7) | 12.9(-15.3) |
| Qwen3-8B | **Ours** | **80.8** | **72.1** | **65.1** | **73.3** | **72.8** |
| Qwen3-8B | + Random Trigger | 74.2(-6.6) | 66.2(-5.9) | 59.4(-5.7) | 67.4(-5.9) | 66.8(-6.0) |
| Qwen3-8B | + Random Anchor | 69.1(-11.7) | 61.4(-10.7) | 53.4(-11.7) | 64.4(-8.9) | 62.1(-10.7) |
| Qwen3-8B | + Random KV Editor | 63.6(-17.2) | 51.5(-20.6) | 41.6(-23.5) | 51.1(-22.2) | 51.9(-20.9) |
| Qwen3-14B | **Ours** | **93.0** | **73.7** | **73.0** | **75.5** | **78.8** |
| Qwen3-14B | + Random Trigger | 87.3(-5.7) | 68.0(-5.7) | 66.8(-6.2) | 69.6(-5.9) | 72.9(-5.9) |
| Qwen3-14B | + Random Anchor | 84.0(-9.0) | 61.8(-11.9) | 64.4(-8.6) | 64.9(-10.6) | 68.8(-10.0) |
| Qwen3-14B | + Random KV Editor | 76.5(-16.5) | 57.3(-16.4) | 54.6(-18.4) | 58.2(-17.3) | 61.7(-17.1) |
| Llama3.1-8B-Instruct | **Ours** | **56.5** | **7.9** | **3.9** | **17.0** | **21.3** |
| Llama3.1-8B-Instruct | + Random Trigger | 50.6(-5.9) | 2.1(-5.8) | 0.0(-3.9) | 10.7(-6.3) | 15.9(-5.4) |
| Llama3.1-8B-Instruct | + Random Anchor | 47.0(-9.5) | 0.0(-7.9) | 0.0(-3.9) | 7.5(-9.5) | 13.6(-7.7) |
| Llama3.1-8B-Instruct | + Random KV Editor | 36.7(-19.8) | 0.0(-7.9) | 0.0(-3.9) | 2.0(-15.0) | 9.7(-11.6) |
| DeepSeek-R1-Distill-Qwen-1.5B | **Ours** | **91.2** | **30.9** | **36.9** | **61.8** | **55.2** |
| DeepSeek-R1-Distill-Qwen-1.5B | + Random Trigger | 84.7(-6.5) | 25.0(-5.9) | 30.3(-6.6) | 55.9(-5.9) | 49.0(-6.2) |
| DeepSeek-R1-Distill-Qwen-1.5B | + Random Anchor | 79.6(-11.6) | 22.5(-8.4) | 27.5(-9.4) | 52.9(-8.9) | 45.6(-9.6) |
| DeepSeek-R1-Distill-Qwen-1.5B | + Random KV Editor | 70.7(-20.5) | 7.6(-23.3) | 11.9(-25.0) | 43.5(-18.3) | 33.4(-21.8) |
| DeepSeek-R1-Distill-Qwen-14B | **Ours** | **94.3** | **65.1** | **41.1** | **88.8** | **72.3** |
| DeepSeek-R1-Distill-Qwen-14B | + Random Trigger | 88.5(-5.8) | 59.4(-5.7) | 34.9(-6.2) | 83.1(-5.7) | 66.5(-5.8) |
| DeepSeek-R1-Distill-Qwen-14B | + Random Anchor | 82.6(-11.7) | 55.7(-9.4) | 32.2(-8.9) | 80.0(-8.8) | 62.6(-9.7) |
| DeepSeek-R1-Distill-Qwen-14B | + Random KV Editor | 69.6(-24.7) | 48.4(-16.7) | 21.0(-20.1) | 71.1(-17.7) | 52.5(-19.8) |
| DeepSeek-R1-Distill-Llama-8B | **Ours** | **91.4** | **47.6** | **35.6** | **87.7** | **65.6** |
| DeepSeek-R1-Distill-Llama-8B | + Random Trigger | 85.6(-5.8) | 41.8(-5.8) | 29.6(-6.0) | 82.0(-5.7) | 59.8(-5.8) |
| DeepSeek-R1-Distill-Llama-8B | + Random Anchor | 79.9(-11.5) | 37.5(-10.1) | 24.3(-11.3) | 79.7(-8.0) | 55.4(-10.2) |
| DeepSeek-R1-Distill-Llama-8B | + Random KV Editor | 73.1(-18.3) | 24.1(-23.5) | 17.3(-18.3) | 66.7(-21.0) | 45.3(-20.3) |

*Table 3.* Intervention Rate (%) of Qwen3 models of different sizes.

| Qwen3 | 0.6B | 1.7B | 4B | 8B | 14B | avg |
|---|---|---|---|---|---|---|
| Intervention Rate | 31.3 | 29.8 | 27.9 | 26.7 | 25.8 | 28.3 |

bone), the average additional latency per sample is below 15%. This demonstrates HEdit's viability for real-world high-efficiency reasoning tasks.

# 5. Conclusion and Discussion

## 5.1. Conclusion

In this paper, we address a fundamental challenge in complex reasoning for Large Language Models: the irreversibility of autoregressive decoding. We introduce HEdit (Hindsight Editing), a novel paradigm that facilitates dynamic intervention in reasoning trajectories. By deconstructing the mechanics of reasoning failures, we identify logical anchors—pivotal nodes in reasoning—and trigger points—moments where cognitive dissonance is captured internally. Utilizing a lightweight editor, HEdit implements a "Sense-Backtrack-Edit-Regenerate" closed-loop mecha-

nism. Empirical results demonstrate that HEdit significantly bolsters the mathematical reasoning performance of models across various scales, requiring a negligible parameter increase of less than 0.5%. Beyond breaking the error cascading effect inherent in autoregressive mechanisms, HEdit provides a fresh perspective on developing reasoning models with endogenous error-correction capabilities and logical reliability.

## 5.2. Limitations and Future Work

While HEdit exhibits exceptional performance in mathematically rigorous tasks with objective certainty, its generalization boundaries warrant further exploration:

- **Ambiguity in Open-ended Domains:** In domains lacking objective verification standards (e.g., creative writing or subjective dialogue), precisely defining the "Trigger" for logical collapse remains a challenge.

- **Complex Multi-error Chains:** The current architecture primarily targets the correction of a single critical anchor. For extremely long reasoning chains with multiple intertwined error sources, future research could

explore multi-round iterative editing or recursive backtracking strategies.

- **The Knowledge Ceiling:** HEdit essentially optimizes the utilization of a model's existing knowledge (error correction) rather than filling intrinsic knowledge gaps (knowledge acquisition). If the base model lacks the underlying knowledge to generate a valid representation at an anchor, the editor's effectiveness will be limited by the model's "knowledge ceiling".

## Impact Statement

This paper presents work whose goal is to advance the field of Machine Learning. There are many potential societal consequences of our work, none which we feel must be specifically highlighted here.

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

# A. Implementation and Training Details

*Table 4.* Layer-wise configurations for HEdit across different backbone models.

| Model | Total Layers ($L$) | Anchor Layer ($L_a$) | Intervention Layer ($L_i$) |
|---|---|---|---|
| Qwen3-0.6B | 28 | 10 | 16 |
| Qwen3-1.7B | 28 | 10 | 16 |
| Qwen3-4B | 36 | 12 | 24 |
| Qwen3-8B | 36 | 12 | 24 |
| Qwen3-14B | 40 | 12 | 30 |
| Llama3.1-8B-Instrcuct | 32 | 10 | 20 |
| DeepSeek-R1-Distill-Qwen-1.5B | 28 | 10 | 16 |
| DeepSeek-R1-Distill-Qwen-7B | 28 | 10 | 16 |
| DeepSeek-R1-Distill-Qwen-14B | 48 | 16 | 36 |
| DeepSeek-R1-Distill-Llama-8B | 32 | 10 | 20 |

The anchor layer $L_a$ is strategically selected at the early-middle stage of the residual stream to capture nascent semantic deviations, while the intervention layer $L_i$ is positioned to apply corrections before the final logit projection. The trigger mechanism calculates the cosine similarity between the hidden states of the $(N-1)$-th and $(N-2)$-th layers, where $N$ denotes the total number of layers. In our paper, the value of k for the top-k selection of anchor and trigger tokens is set to 10.

*Table 5.* Hyperparameters and training configurations for the MLP-based KV Editor.

| Hyperparameter | Value |
|---|---|
| Hidden Dimension 1 | 2048 |
| Hidden Dimension 2 | 1024 |
| Activation Function | ReLU |
| Dropout Rate | 0.1 |
| Batch Size | 16 |
| Learning Rate | 0.0001 |
| Weight Decay | 0.00001 |
| Training Epochs | 100 |
| Train/Val Split | 0.8 / 0.2 |

To ensure the robustness of the KV Editor, we employ a standard MLP architecture with a bottleneck structure (2048 $\rightarrow$ 1024). The model is trained using the AdamW optimizer with a weight decay of 1e-5 to prevent overfitting on the 1k GSM8K trajectory samples.

# B. Main Results Details

*Table 6.* Detailed results of all models and methods.

| Model | Method | MATH500 | AIME24 | AIME25 | AMC23 | Average |
|---|---|---|---|---|---|---|
| Qwen3-0.6B | Original | 43.2 | 4.3 | 4.6 | 17.9 | 17.5 |
| Qwen3-0.6B | + CoT | 45.5(+2.3) | 7.1(+2.8) | 6.0(+1.4) | 19.2(+1.3) | 19.4(+1.9) |
| Qwen3-0.6B | +Naive BT | 43.6(+0.4) | 5.7(+1.4) | 4.9(+0.3) | 18.4(+0.5) | 18.1(+0.6) |
| Qwen3-0.6B | **+ Ours** | **56.1(+12.9)** | **11.1(+6.8)** | **16.5(+11.9)** | **29.3(+11.4)** | **28.2(+10.8)** |
| Qwen3-1.7B | Original | 65.2 | 10.4 | 13.1 | 47.4 | 34.0 |
| Qwen3-1.7B | + CoT | 67.4(+2.2) | 12.5(+2.1) | 15.8(+2.7) | 49.5(+2.1) | 36.3(+2.3) |
| Qwen3-1.7B | +Naive BT | 67.3(+2.1) | 11.1(+0.7) | 15.5(+2.4) | 48.2(+0.9) | 35.5(+1.5) |
| Qwen3-1.7B | **+ Ours** | **75.5(+10.3)** | **18.3(+7.9)** | **24.1(+11.0)** | **57.9(+10.5)** | **44.0(+9.9)** |
| Qwen3-4B | Original | 67.3 | 43.2 | 40.4 | 58.2 | 52.3 |
| Qwen3-4B | + CoT | 69.3(+2.0) | 48.0(+4.8) | 41.9(+1.5) | 59.5(+1.3) | 54.7(+2.4) |
| Qwen3-4B | +Naive BT | 67.4(+0.1) | 45.6(+2.4) | 41.9(+1.5) | 60.1(+1.9) | 53.8(+1.5) |
| Qwen3-4B | **+ Ours** | **77.3(+10.0)** | **48.8(+5.6)** | **45.0(+4.6)** | **64.8(+6.6)** | **59.0(+6.7)** |
| Qwen3-8B | Original | 73.7 | 65.6 | 61.3 | 65.3 | 66.5 |
| Qwen3-8B | + CoT | 74.8(+1.1) | 70.5(+4.9) | 64.7(+3.4) | 69.8(+4.5) | 70.0(+3.5) |
| Qwen3-8B | +Naive BT | 74.1(+0.4) | 67.4(+1.8) | 61.6(+0.3) | 67.7(+2.4) | 67.7(+1.2) |
| Qwen3-8B | **+ Ours** | **80.8(+7.1)** | **72.1(+6.5)** | **65.1(+3.8)** | **73.3(+8.0)** | **72.8(+6.3)** |
| Qwen3-14B | Original | 86.6 | 68.4 | 66.7 | 70.2 | 73.0 |
| Qwen3-14B | + CoT | 91.5(+4.9) | 69.6(+1.2) | 70.9(+4.2) | 74.4(+4.2) | 76.6(+3.6) |
| Qwen3-14B | +Naive BT | 89.3(+2.7) | 70.4(+2.0) | 67.2(+0.5) | 72.5(+2.3) | 74.8(+1.9) |
| Qwen3-14B | **+ Ours** | **93.0(+6.4)** | **73.7(+5.3)** | **73.0(+6.3)** | **75.5(+5.3)** | **78.8(+5.8)** |
| Llama3.1-8B-Instruct | Original | 47.6 | 5.1 | 1.9 | 8.6 | 15.8 |
| Llama3.1-8B-Instruct | + CoT | 49.1(+1.5) | 7.1(+2.0) | 2.7(+0.8) | 11.3(+2.7) | 17.6(+1.8) |
| Llama3.1-8B-Instruct | + Naive BT | 48.8(+1.2) | 7.0(+1.9) | 2.0(+0.1) | 10.9(+2.3) | 17.2(+1.4) |
| Llama3.1-8B-Instruct | **+ Ours** | **56.5(+8.9)** | **7.9(+2.8)** | **3.9(+2.0)** | **17.0(+8.4)** | **21.3(+5.5)** |
| DeepSeek-R1-Distill-Qwen-1.5B | Original | 83.2 | 27.5 | 25.4 | 55.4 | 47.9 |
| DeepSeek-R1-Distill-Qwen-1.5B | + CoT | 85.0(+1.8) | 28.8(+1.3) | 28.1(+2.7) | 58.5(+3.1) | 50.1(+2.2) |
| DeepSeek-R1-Distill-Qwen-1.5B | +Naive BT | 84.2(+1.0) | 29.8(+2.3) | 26.6(+1.2) | 58.0(+2.6) | 49.6(+1.8) |
| DeepSeek-R1-Distill-Qwen-1.5B | **+ Ours** | **91.2(+8.0)** | **30.9(+3.4)** | **36.9(+11.5)** | **61.8(+6.4)** | **55.2(+7.3)** |
| DeepSeek-R1-Distill-Qwen-7B | Original | 92.2 | 51.6 | 36.9 | 81.0 | 65.4 |
| DeepSeek-R1-Distill-Qwen-7B | + CoT | 93.2(+1.0) | 55.7(+4.1) | 41.5(+4.6) | 84.5(+3.5) | 68.7(+3.3) |
| DeepSeek-R1-Distill-Qwen-7B | +Naive BT | 93.0(+0.8) | 53.3(+1.7) | 37.7(+0.9) | 83.4(+2.4) | 66.8(+1.4) |
| DeepSeek-R1-Distill-Qwen-7B | **+ Ours** | **93.3(+1.1)** | **55.8(+4.2)** | **43.2(+6.4)** | **87.9(+6.9)** | **70.1(+4.7)** |
| DeepSeek-R1-Distill-Qwen-14B | Original | 93.1 | 62.7 | 39.5 | 85.2 | 70.1 |
| DeepSeek-R1-Distill-Qwen-14B | + CoT | 94.0(+0.9) | 64.7(+2.0) | **44.2(+4.7)** | 88.7(+3.5) | **72.8(+2.6)** |
| DeepSeek-R1-Distill-Qwen-14B | +Naive BT | 93.7(+0.6) | 63.8(+1.1) | 40.3(+0.8) | 86.2(+1.0) | 71.0(+0.9) |
| DeepSeek-R1-Distill-Qwen-14B | **+ Ours** | **94.3(+1.2)** | **65.1(+2.4)** | 41.1(+1.6) | **88.8(+3.6)** | 72.3(+2.2) |
| DeepSeek-R1-Distill-Llama-8B | Original | 89.7 | 44.8 | 30.8 | 79.2 | 61.1 |
| DeepSeek-R1-Distill-Llama-8B | + CoT | 91.3(+1.6) | 47.3(+2.5) | **35.7(+4.9)** | 83.9(+4.7) | 64.6(+3.4) |
| DeepSeek-R1-Distill-Llama-8B | +Naive BT | 91.1(+1.4) | 45.7(+0.9) | 32.0(+1.2) | 82.1(+2.9) | 62.7(+1.6) |
| DeepSeek-R1-Distill-Llama-8B | **+ Ours** | **91.4(+1.7)** | **47.6(+2.8)** | 35.6(+4.8) | **87.7(+8.5)** | **65.6(+4.5)** |

# C. Ablation study Details

*Table 7.* Ablation study results.

| Model | Method | MATH500 | AIME24 | AIME25 | AMC23 | Average |
|---|---|---|---|---|---|---|
| Qwen3-0.6B | **Ours** | **56.1** | **11.1** | **16.5** | **29.3** | **28.2** |
| Qwen3-0.6B | + Random Trigger | 49.7(-6.4) | 5.6(-5.5) | 10.6(-5.9) | 22.8(-6.5) | 22.2(-6.0) |
| Qwen3-0.6B | + Random Anchor | 46.1(-10.0) | 2.8(-8.3) | 5.2(-11.3) | 18.8(-10.5) | 18.2(-10.0) |
| Qwen3-0.6B | + Random KV Editor | 40.0(-16.1) | 0.0(-11.1) | 0.0(-16.5) | 11.6(-17.7) | 12.9(-15.3) |
| Qwen3-1.7B | **Ours** | **75.5** | **18.3** | **24.1** | **57.9** | **44.0** |
| Qwen3-1.7B | + Random Trigger | 69.2(-6.3) | 12.0(-6.3) | 17.6(-6.5) | 51.5(-6.4) | 37.6(-6.4) |
| Qwen3-1.7B | + Random Anchor | 65.0(-10.5) | 8.4(-9.9) | 14.9(-9.2) | 49.9(-8.0) | 34.6(-9.4) |
| Qwen3-1.7B | + Random KV Editor | 57.4(-18.1) | 0.0(-18.3) | 2.1(-22.0) | 39.1(-18.8) | 24.6(-19.4) |
| Qwen3-4B | **Ours** | **77.3** | **48.8** | **45.0** | **64.8** | **59.0** |
| Qwen3-4B | + Random Trigger | 70.8(-6.5) | 43.1(-5.7) | 38.6(-6.4) | 58.4(-6.4) | 52.7(-6.3) |
| Qwen3-4B | + Random Anchor | 66.4(-10.9) | 38.6(-10.2) | 34.8(-10.2) | 55.5(-9.3) | 48.8(-10.2) |
| Qwen3-4B | + Random KV Editor | 59.1(-18.2) | 30.5(-18.3) | 21.9(-23.1) | 44.4(-20.4) | 39.0(-20.0) |
| Qwen3-8B | **Ours** | **80.8** | **72.1** | **65.1** | **73.3** | **72.8** |
| Qwen3-8B | + Random Trigger | 74.2(-6.6) | 66.2(-5.9) | 59.4(-5.7) | 67.4(-5.9) | 66.8(-6.0) |
| Qwen3-8B | + Random Anchor | 69.1(-11.7) | 61.4(-10.7) | 53.4(-11.7) | 64.4(-8.9) | 62.1(-10.7) |
| Qwen3-8B | + Random KV Editor | 63.6(-17.2) | 51.5(-20.6) | 41.6(-23.5) | 51.1(-22.2) | 51.9(-20.9) |
| Qwen3-14B | **Ours** | **93.0** | **73.7** | **73.0** | **75.5** | **78.8** |
| Qwen3-14B | + Random Trigger | 87.3(-5.7) | 68.0(-5.7) | 66.8(-6.2) | 69.6(-5.9) | 72.9(-5.9) |
| Qwen3-14B | + Random Anchor | 84.0(-9.0) | 61.8(-11.9) | 64.4(-8.6) | 64.9(-10.6) | 68.8(-10.0) |
| Qwen3-14B | + Random KV Editor | 76.5(-16.5) | 57.3(-16.4) | 54.6(-18.4) | 58.2(-17.3) | 61.7(-17.1) |
| Llama3.1-8B-Instruct | **Ours** | **56.5** | **7.9** | **3.9** | **17.0** | **21.3** |
| Llama3.1-8B-Instruct | + Random Trigger | 50.6(-5.9) | 2.1(-5.8) | 0.0(-3.9) | 10.7(-6.3) | 15.9(-5.4) |
| Llama3.1-8B-Instruct | + Random Anchor | 47.0(-9.5) | 0.0(-7.9) | 0.0(-3.9) | 7.5(-9.5) | 13.6(-7.7) |
| Llama3.1-8B-Instruct | + Random KV Editor | 36.7(-19.8) | 0.0(-7.9) | 0.0(-3.9) | 2.0(-15.0) | 9.7(-11.6) |
| DeepSeek-R1-Distill-Qwen-1.5B | **Ours** | **91.2** | **30.9** | **36.9** | **61.8** | **55.2** |
| DeepSeek-R1-Distill-Qwen-1.5B | + Random Trigger | 84.7(-6.5) | 25.0(-5.9) | 30.3(-6.6) | 55.9(-5.9) | 49.0(-6.2) |
| DeepSeek-R1-Distill-Qwen-1.5B | + Random Anchor | 79.6(-11.6) | 22.5(-8.4) | 27.5(-9.4) | 52.9(-8.9) | 45.6(-9.6) |
| DeepSeek-R1-Distill-Qwen-1.5B | + Random KV Editor | 70.7(-20.5) | 7.6(-23.3) | 11.9(-25.0) | 43.5(-18.3) | 33.4(-21.8) |
| DeepSeek-R1-Distill-Qwen-7B | **Ours** | **93.3** | **55.8** | **43.2** | **87.9** | **70.1** |
| DeepSeek-R1-Distill-Qwen-7B | + Random Trigger | 87.4(-5.9) | 49.6(-6.2) | 37.5(-5.7) | 82.1(-5.8) | 64.2(-5.9) |
| DeepSeek-R1-Distill-Qwen-7B | + Random Anchor | 81.7(-11.6) | 43.9(-11.9) | 34.3(-8.9) | 76.3(-11.6) | 59.0(-11.1) |
| DeepSeek-R1-Distill-Qwen-7B | + Random KV Editor | 71.1(-22.2) | 37.0(-18.8) | 23.9(-19.3) | 66.0(-21.9) | 49.5(-20.6) |
| DeepSeek-R1-Distill-Qwen-14B | **Ours** | **94.3** | **65.1** | **41.1** | **88.8** | **72.3** |
| DeepSeek-R1-Distill-Qwen-14B | + Random Trigger | 88.5(-5.8) | 59.4(-5.7) | 34.9(-6.2) | 83.1(-5.7) | 66.5(-5.8) |
| DeepSeek-R1-Distill-Qwen-14B | + Random Anchor | 82.6(-11.7) | 55.7(-9.4) | 32.2(-8.9) | 80.0(-8.8) | 62.6(-9.7) |
| DeepSeek-R1-Distill-Qwen-14B | + Random KV Editor | 69.6(-24.7) | 48.4(-16.7) | 21.0(-20.1) | 71.1(-17.7) | 52.5(-19.8) |
| DeepSeek-R1-Distill-Llama-8B | **Ours** | **91.4** | **47.6** | **35.6** | **87.7** | **65.6** |
| DeepSeek-R1-Distill-Llama-8B | + Random Trigger | 85.6(-5.8) | 41.8(-5.8) | 29.6(-6.0) | 82.0(-5.7) | 59.8(-5.8) |
| DeepSeek-R1-Distill-Llama-8B | + Random Anchor | 79.9(-11.5) | 37.5(-10.1) | 24.3(-11.3) | 79.7(-8.0) | 55.4(-10.2) |
| DeepSeek-R1-Distill-Llama-8B | + Random KV Editor | 73.1(-18.3) | 24.1(-23.5) | 17.3(-18.3) | 66.7(-21.0) | 45.3(-20.3) |

