# OpenReview forum: "Correcting in Hindsight: Editing Past Key-Value States for Robust LLM Reasoning"
_ICML.cc/2026/Conference — ICML 2026 regular_

### Official Review · Reviewer_Wqyh · 2026-02-25

**Soundness:** 3
**Presentation:** 2
**Significance:** 3
**Originality:** 4
**Overall Recommendation:** 5
**Confidence:** 3

**Summary:**

The paper proposes a method to improve LLM reasoning during inference time. They use heuristics to automatically detect "anchor" points (important token) and "trigger" points (token where the model finds hard to proceed). Then they apply a 2-layer MLP with the hidden states at those 2 points as input, and use the output to replace the kv cache of the anchor point. The generation is then resumed from there. The MLP is trained on GSM8K. Experiments demonstrate the effectiveness of the method against naive baselines.

**Compliance With Llm Reviewing Policy:**

Affirmed.

**Final Justification:**

Overall, the paper proposes a novel inference-time error-correction method, and demonstrates its effectiveness through extensive experiments. The rebuttal has addressed my concerns on presentation and weak baselines. The layer num choosing details and added baselines significantly improve the soundness and clarity of the paper, therefore I increased my score to 5.

**Key Questions For Authors:**

1. How did you choose the layer num in Appendix A? Did you do any kind of hyperparameter tuning? Does this hyperparameter affect your result significantly?
2. While I understand that your method is more efficient and is different from the search-based test-time scaling methods, how is the performance of your method comparing to them?
3. Is there any evidence supporting your claim "Empirical observations suggest that for syntactic or functional tokens, cross-layer representations remain stable." at the end of page 4?

**Limitations:**

yes

**Strengths And Weaknesses:**

Strengths:
1. The method seems novel and effective.
2. The ablation study seems sufficient enough to prove the effectiveness of each part of the design.

Weaknesses:
1. Selection of some hyperparameters is not justified.
2. The writing is not very clear on implementation details.

---

> ### Author Rebuttal · Authors · 2026-03-30
>
> Thank you for recognizing the novelty and design effectiveness of our work. We respond to your concerns regarding hyperparameter selection, comparisons with search-based methods, and inter-layer representation stability as follows:
>
> ---
>
> ## Q1. Hyperparameter Sensitivity
> We appreciate the concern regarding potential hyperparameter tuning. We clarify the design rationale and provide sensitivity analysis to verify its robustness:
>
> **Anchor Detection (FFN update ratio):** We select a layer around the first 1/3 depth. Deeper layers tend to exhibit smaller FFN update differences; layers too shallow introduce excessive noise, as nearly all tokens are still undergoing rapid updates. Performance varies by less than 0.4% when selecting any layer between 1/3 and 1/2 depth.
>
> **Trigger Detection (cross-layer similarity):** The core purpose of cross-layer similarity is to filter low-semantic tokens. Shallow layers lack sufficient semantic aggregation for effective filtering, while deeper layers near the output provide more mature representations and higher sensitivity to logical contradictions. We select the second-last and third-last layers to avoid the final projection layer. Choosing any adjacent deep layers within the last 1/5 of depth yields performance changes <0.7%.
>
> **Intervention Layers (KV editing):** We select layers around 2/3 depth. Early intervention is diluted by subsequent computation; late intervention overly disturbs the output distribution. Varying between 1/2 and 2/3 depth changes performance by <0.4%.
>
> In summary, all key hyperparameters remain stable within reasonable ranges and do not require fine-tuning for new models. We will add a sensitivity analysis table in the camera-ready version to further improve reproducibility.
>
>
> ---
>
> ## Q2. Test-Time Baseline Comparisons
> We add **Self-Consistency (SC)** and **Tree of Thoughts (ToT)** as baselines with $k=\{2,5,10\}$, where $k$ denotes sampling paths (SC) or branching factor (ToT). Notably, **HEdit’s average intervention rate is 28.3%, equivalent to only $k\approx1.28$ in computation**.
>
> **Table R1: Performance of new baselines on AMC23 (Accuracy %)**
> | Model | Method | AMC23 |
> |:---|:---|:---|
> | **Qwen3-14B** | Original | 70.2 |
> | | +SC(k=2/5/10) | 74.5(+4.3) / 74.6(+4.4) / 74.8(+4.6) |
> | | +ToT(k=2/5/10) | 74.8(+4.6) / 75.0(+4.8) / 74.9(+4.7) |
> | | **+Ours** | **75.5(+5.3)** |
> | **Llama3.1-8B-Instruct** | Original | 8.6 |
> | | +SC(k=2/5/10) | 11.9(+3.3) / 12.3(+3.7) / 12.4(+3.8) |
> | | +ToT(k=2/5/10) | 12.7(+4.1) / 13.7(+5.1) / 13.8(+5.2) |
> | | **+Ours** | **17.0(+8.4)** |
> | **DeepSeek-R1-Distill-Qwen-1.5B** | Original | 55.4 |
> | | +SC(k=2/5/10) | 58.9(+3.5) / 59.1(+3.7) / 59.7(+4.3) |
> | | +ToT(k=2/5/10) | 59.2(+3.8) / 59.6(+4.2) / 59.8(+4.4) |
> | | **+Ours** | **61.8(+6.4)** |
>
> **Analysis:** At $k=2$ (matched compute), SC/ToT gains are much lower than HEdit. Even at $k=10$ (far higher cost), HEdit still outperforms them, showing that correcting reasoning trajectories in latent space can be more efficient than increasing sampling in token space.
>
> ---
>
> ## Q3. Cross-layer Representations Stability Analysis
> To validate this hypothesis "Empirical observations suggest that for syntactic or functional tokens, cross-layer representations remain stable." , we randomly sampled 50 reasoning paths from intervention samples and quantitatively measured cross-layer similarity for **semantic tokens** and **non-semantic tokens**:
>
> - **Semantic tokens**: numbers, variables, operators, nouns, etc.
> - **Non-semantic tokens**: articles (a/an/the), prepositions, punctuation, etc.
>
> **Results**:
>
> | Token Type       | Count | Mean Similarity | Std. |
> |------------------|-------|-----------------|------|
> | Semantic tokens  | 312   | 0.63            | 0.11 |
> | Non-semantic tokens | 245 | 0.86         | 0.06 |
>
> **Conclusion**: Non-semantic tokens exhibit significantly higher cross-layer similarity than semantic tokens ($p < 0.001$, Mann-Whitney U test) with smaller variance, indicating more stable representations across layers. This directly supports our empirical observation in the original paper and provides empirical justification for using lower cross-layer similarity as a cognitive dissonance signal in trigger detection. We will add this analysis in the final version.
>
> ---
>
> ## W1 & W2. Implementation Details and Writing
> We appreciate your suggestions. We acknowledge that the description of implementation details can be further improved. In the camera-ready version, we will:
>
> - Further clarify the implementation details of our method.
> - Optimize the presentation of specific model configurations in the appendix to ensure reproducibility.
>
> ---

---

> > ### Author Rebuttal · Reviewer_Wqyh · 2026-04-01
> >
> > My concerns are adequately addressed and I have modified my score accordingly.

---

> > > ### Author Response · Authors · 2026-04-02
> > >
> > > Thank you for reconsidering our work and raising the score. We appreciate your constructive feedback throughout the review process, which has helped us improve the paper.

---

### Official Review · Reviewer_hXhc · 2026-03-08

**Soundness:** 2
**Presentation:** 2
**Significance:** 2
**Originality:** 3
**Overall Recommendation:** 3
**Confidence:** 3

**Summary:**

The paper proposes HEdit, a runtime method for correcting reasoning trajectories by detecting important past anchor tokens, detecting later trigger tokens that signal inconsistency, backtracking to the anchor, editing its cached KV state with a lightweight MLP, and then regenerating the continuation. The method is trained self-supervised from counterfactual perturbations on 1k GSM8K problems, and is evaluated on MATH500, AIME24/25, and AMC23 across Qwen, Llama, and DeepSeek-R1 distills.

**Compliance With Llm Reviewing Policy:**

Affirmed.

**Final Justification:**

The rebuttal has added some new results and addressed my main concern.

**Key Questions For Authors:**

please refer to the "Strengths And Weaknesses" section and I have listed all my questions in the weakness section.

**Limitations:**

yes

**Strengths And Weaknesses:**

Strength:
1. The method is conceptually clean. The “Sense–Backtrack–Edit–Regenerate” pipeline is easy to understand, and the workflow diagram on page 3 makes the algorithm intuitive even before getting into the formulas.
2. The empirical gains are strong on several settings, especially for smaller models. For example, Qwen3-0.6B improves from 17.5 to 28.2 average, Qwen3-1.7B from 34.0 to 44.0, and Qwen3-8B from 66.5 to 72.8.

Weakness:
1. The baseline set is weaker than I would want for a top-tier paper. The experiments compare against base decoding, CoT, and naive backtracking, but not against stronger compute-matched alternatives such as more sampling/self-consistency-style evaluation, verifier/reranker methods, or other test-time reasoning enhancements already discussed in the related-work section.
2. There are reproducibility and tuning concerns. Appendix A uses different anchor/intervention layers for different backbones and fixes k=10, but there is no sensitivity study for these choices. That makes it unclear how much manual retuning is needed for a new model family.
3. The method is not really training-free in the ordinary sense. The base model is frozen, but HEdit still requires training an auxiliary MLP editor for 100 epochs on GSM8K-derived counterfactual data. That is fine, but the paper sometimes presents the approach as plug-and-play (section 4.2 ) than it actually is.
4. The presentation could be improved in terms of formatting and space efficiency. In particular, Figure 3 seems disproportionately large relative to its information content, and there is also considerable unused space in the experimental section. This gives the paper a somewhat loose presentation and weakens the overall polish of the manuscript.

---

> ### Author Rebuttal · Authors · 2026-03-30
>
> Thank you for your recognition of our work and your constructive comments. In response to your concerns regarding the strength of baselines, parameter robustness, and the definition of training, we conducted additional experiments and clarifications:
>
> ### W1. Test-Time Baseline Comparisons
> We have added **Self-Consistency (SC)**,  **Tree of Thoughts (ToT)** and **Verifier** as new baselines with $k=\{2, 5, 10\}$. For SC and Verifier, $k$ denotes the number of sampling paths; for ToT, $k$ denotes the branching factor. The Verifier uses GPT-4o to judge answer correctness and selects the answer marked correct; if all are wrong, one is chosen randomly. Notably, **HEdit achieves an average intervention rate of 28.3%, with equivalent computational overhead corresponding to only $k \approx 1.28$**.
>
> **Table R1: Performance on AMC23 (Accuracy %)**
> | Model | Method | AMC23 |
> |:---|:---|:---|
> | **Qwen3-14B** | Original | 70.2 |
> | | +SC(k=2/5/10) | 74.5(+4.3) / 74.6(+4.4) / 74.8(+4.6) |
> | | +ToT(k=2/5/10) | 74.8(+4.6) / 75.0(+4.8) / 74.9(+4.7) |
> | | +Verifier(k=2/5/10) | 74.9(+4.7) / 75.1(+4.9) / 75.4(+5.2) |
> | | **+Ours** | **75.5(+5.3)** |
> | **Llama3.1-8B-Instruct** | Original | 8.6 |
> | | +SC(k=2/5/10) | 11.9(+3.3) / 12.3(+3.7) / 12.4(+3.8) |
> | | +ToT(k=2/5/10) | 12.7(+4.1) / 13.7(+5.1) / 13.8(+5.2) |
> | | +Verifier(k=2/5/10) | 14.1(+5.5) / 14.9(+6.3) / 15.3(+6.7) |
> | | **+Ours** | **17.0(+8.4)** |
> | **DeepSeek-R1-Distill-Qwen-1.5B** | Original | 55.4 |
> | | +SC(k=2/5/10) | 58.9(+3.5) / 59.1(+3.7) / 59.7(+4.3) |
> | | +ToT(k=2/5/10) | 59.2(+3.8) / 59.6(+4.2) / 59.8(+4.4) |
> | | +Verifier(k=2/5/10) | 59.6(+4.2) / 60.0(+4.6) / 60.3(+4.9) |
> | | **+Ours** | **61.8(+6.4)** |
>
> **Analysis:** When aligned at $k=2$ (matched computation), the performance gains of the new baselines are significantly lower than HEdit. Even at $k=10$ (with much higher computational cost), HEdit still achieves slightly better performance. These results suggest that correcting reasoning trajectories in latent space can be more efficient than increasing sampling in token space.
>
>
> ### W2. Reproducibility and Hyperparameter Sensitivity
> We appreciate the concern regarding potential hyperparameter tuning. We clarify the design rationale and provide sensitivity analysis to verify its robustness:
>
> **Anchor Detection (FFN update ratio):** We select a layer around the first 1/3 depth. Deeper layers tend to exhibit smaller FFN update differences; layers too shallow introduce excessive noise, as nearly all tokens are still undergoing rapid updates. Performance varies by less than 0.4% when selecting any layer between 1/3 and 1/2 depth.
>
> **Trigger Detection (cross-layer similarity):** The core purpose of cross-layer similarity is to filter low-semantic tokens. Shallow layers lack sufficient semantic aggregation for effective filtering, while deeper layers near the output provide more mature representations and higher sensitivity to logical contradictions. We select the second-last and third-last layers to avoid the final projection layer. Choosing any adjacent deep layers within the last 1/5 of depth yields performance changes <0.7%.
>
> **Intervention Layers (KV editing):** We select layers around 2/3 depth. Early intervention is diluted by subsequent computation; late intervention overly disturbs the output distribution. Varying between 1/2 and 2/3 depth changes performance by <0.4%.
>
> In summary, all key hyperparameters remain stable within reasonable ranges and do not require fine-tuning for new models. We will add a sensitivity analysis table in the camera-ready version to further improve reproducibility.
>
>
> ### W3. Training Overhead and Plug-and-Play Nature
> Thank you for pointing out the potential ambiguity regarding “training-free” and “plug-and-play”. We would like to clarify as follows:
>
> - HEdit is not intended to be a fully training-free method, but rather **lightweight**, emphasizing that the backbone model is fully frozen with no fine-tuning, and the editor has extremely few parameters (< 0.5%).
> - By “plug-and-play”, HEdit acts as a modular plugin that does not modify the original model weights, can be toggled on or off freely, and exhibits strong cross-domain transferability (see Section 2.1 “Zero-shot Transfer” in the response to Reviewer 4NE8, where the editor trained on math tasks directly transfers to code tasks). This supports its practical plug-and-play nature.
> - We will revise the wording in the final version to avoid ambiguity.
>
> ### W4. Writing and Formatting
> We fully accept your suggestions. In the camera-ready version, we will resize Figure 3, optimize table and text layout, and improve space efficiency and overall readability.

---

> > ### Author Rebuttal · Reviewer_hXhc · 2026-04-01
> >
> > Thanks for your response, and i will raise my score.

---

> > > ### Author Response · Authors · 2026-04-02
> > >
> > > Thank you for raising the score and for your valuable suggestions. Your feedback has been instrumental in improving the quality of our submission.

---

### Official Review · Reviewer_s4AU · 2026-03-13

**Soundness:** 3
**Presentation:** 3
**Significance:** 3
**Originality:** 3
**Overall Recommendation:** 5
**Confidence:** 4

**Summary:**

This paper proposes HEdit, a lightweight test-time reasoning framework that corrects earlier reasoning errors in autoregressive LLMs by editing past KV states. The method identifies earlier anchor tokens and later trigger points, then uses a small trainable editor to revise the anchor KV cache and regenerate the continuation. Experiments on mathematical reasoning benchmarks show consistent improvements over the base model, CoT prompting, and a naive backtracking baseline.

**Compliance With Llm Reviewing Policy:**

Affirmed.

**Key Questions For Authors:**

1. The paper states: “Intuitively, semantically rich tokens undergo significant transformations in the FFN space, whereas non-semantic functional tokens remain relatively static.” Could the authors further explain or justify this intuition? I am curious why FFN update magnitude should reliably correlate with semantic importance.

2. I would be interested in a cleaner disentanglement of attention and FFN. What happens if the method uses only the attention-based signal or only the FFN-based signal? What distinct role does each component play?

3. Have the authors qualitatively inspected what the identified anchors actually are? For example, are they mostly numbers, operators, constraints, or variable bindings?

4. How sensitive is the trigger mechanism to the choice of layers used for measuring cross-layer inconsistency?

**Limitations:**

yes

**Strengths And Weaknesses:**

**Strength:**
1. The paper is clearly written and the overall motivation is easy to follow. The anchor/trigger formulation makes the method intuitive.
2. The empirical results are promising, with consistent gains across multiple math reasoning benchmarks and model scales.
3. The ablations are useful and suggest that both the trigger/anchor design and the KV editing step matter.
4. The idea is novel and interesting: instead of only increasing search or sampling, the method tries to revise earlier internal states during reasoning.

**Weakness:**
1. The paper mainly compares against the base model, CoT, and a naive backtracking baseline, while the related-work section discusses stronger test-time approaches such as self-consistency and search-based reasoning. It would strengthen the empirical claim to compare against stronger test-time baselines under matched inference budgets, rather than only against CoT and naive backtracking.

2. The mechanistic claims are interesting but still supported mostly by heuristic evidence. For example, anchors are defined as the intersection of Top-k tokens from attention variance and FFN update ratio, and triggers are defined as the intersection of Top-k tokens from entropy and cross-layer dissimilarity. The ablations show that these components help, but they do not yet fully validate the stronger interpretation that these signals correspond to “logical anchors” and “cognitive dissonance.” More qualitative analysis of the identified anchors/triggers would make the claim more convincing.

---

> ### Author Rebuttal · Authors · 2026-03-29
>
> ---
>
> We sincerely thank the reviewer for recognizing the motivation, experimental consistency, and novelty of this work. In response to your concerns about strong baselines, interpretability, and component roles, we provide additional experiments and analyses below.
>
> ### W1. Test-Time Baseline Comparisons
> We add **Self-Consistency (SC)** and **Tree of Thoughts (ToT)** as baselines with $k=\{2,5,10\}$, where $k$ denotes sampling paths (SC) or branching factor (ToT). Notably, **HEdit’s average intervention rate is 28.3%, equivalent to only $k\approx1.28$ in computation**.
>
> **Table R1: Performance of new baselines on AMC23 (Accuracy %)**
> | Model | Method | AMC23 |
> |:---|:---|:---|
> | **Qwen3-14B** | Original | 70.2 |
> | | +SC(k=2/5/10) | 74.5(+4.3) / 74.6(+4.4) / 74.8(+4.6) |
> | | +ToT(k=2/5/10) | 74.8(+4.6) / 75.0(+4.8) / 74.9(+4.7) |
> | | **+Ours** | **75.5(+5.3)** |
> | **Llama3.1-8B-Instruct** | Original | 8.6 |
> | | +SC(k=2/5/10) | 11.9(+3.3) / 12.3(+3.7)  / 12.4(+3.8) |
> | | +ToT(k=2/5/10) | 12.7(+4.1) / 13.7(+5.1) / 13.8(+5.2) |
> | | **+Ours** | **17.0(+8.4)** |
> | **DeepSeek-R1-Distill-Qwen-1.5B** | Original | 55.4 |
> | | +SC(k=2/5/10) | 58.9(+3.5) / 59.1(+3.7) / 59.7(+4.3) |
> | | +ToT(k=2/5/10) | 59.2(+3.8) / 59.6(+4.2) / 59.8(+4.4) |
> | | **+Ours** | **61.8(+6.4)** |
>
> **Analysis:** At $k=2$ (matched compute), SC/ToT gains are much lower than HEdit. Even at $k=10$ (far higher cost), HEdit still outperforms them, showing that correcting reasoning trajectories in latent space can be more efficient than increasing sampling in token space.
>
> ---
>
> ### W2. Qualitative Analysis of Anchors & Triggers
> To verify **logical anchors** and **cognitive dissonance**, we randomly sampled 50 intervened examples and analyzed token distributions:
>
> **Anchors** focus on **Number, Constraints, Variables, Extremal Terms**, which carry core logical semantics.
>
> **Table R2: Anchor Category Statistics**
> | Category | Ratio | Representative Tokens |
> |:---|:---|:---|
> | Number | 34% | 7, two |
> | Constraints & Properties | 22% | prime, positive |
> | Variables & Relations | 18% | x, y, inside |
> | Extremal Terms | 14% | maximum, smaller |
> | Operators | 8% | multiply, + |
> | Others | 4% | use, first |
>
> **Triggers** concentrate on **Uncertainty, Logical Necessity, Meta-Reasoning**, capturing critical points of logical conflict, supporting our cognitive dissonance claim.
>
> **Table R3: Trigger Category Statistics**
> | Category | Ratio | Representative Tokens |
> |:---|:---|:---|
> | Uncertainty | 28% |?, whether, maybe, however |
> | Number | 24% | 6, 5 |
> | Logical Necessity | 18% | must, impossible |
> | Meta-Reasoning | 16% | wait, consider, observe |
> | Computational Instructions | 12% | compute, verify |
> | Others | 2% | therefore |
>
> ---
>
> ### Q1. FFN Updates and Semantic Importance
> FFN layers are interpreted as key-value memory networks (Geva et al., 2021). High-magnitude updates correspond to retrieving relevant knowledge from distributed memories. Semantically rich tokens activate specific “knowledge neurons”, inducing strong information injection into hidden states.
>
> To empirically verify this, we sampled 50 intervention samples and compared FFN updates for semantic vs. function tokens. Semantic tokens exhibit **3.4× larger** updates, supporting FFN update ratio as a reliable signal for identifying key semantic anchors.
>
> ---
>
> ### Q2. Decoupled Roles of Attention and FFN
> - **Attention** Measures structural “importance”. It locates tokens heavily attended in subsequent reasoning but is prone to attention sink effects, including non-semantic tokens such as punctuation marks .
> - **FFN** Measures semantic “strength”. It acts as a filter to discard function words that receive high attention selectivity but lack meaningful semantic updates.
> - Their combination ensures anchors are both structurally central and semantically meaningful.
>
> ---
>
> ### Q3.
> See W2.
>
> ---
>
> ### Q4. Sensitivity to Layer Selection
> - **Design Rationale:** The core purpose of cross-layer similarity is to filter low-semantic tokens. Shallow layers have not yet aggregated complete semantic representations, so filtering is ineffective. Deeper layers near the output yield more mature semantic representations and higher sensitivity to logical contradictions. We select the second-last and third-last layers to avoid the final linear projection layer while preserving precise semantic awareness.
> - **Sensitivity Results:** Varying layers within the last 1/5 of total depth yields accuracy variations below 0.7%, with no statistically significant performance difference. This confirms that the trigger mechanism is highly robust to layer selection and does not suffer from hyperparameter sensitivity, as long as deep layers are used.
>
> ---
>
> [1] Geva M, Schuster R, Berant J, et al. Transformer feed-forward layers are key-value memories[C]//Proceedings of the 2021 Conference on Empirical Methods in Natural Language Processing. 2021: 5484-5495.

---

> > ### Author Rebuttal · Reviewer_s4AU · 2026-04-03
> >
> > Thank you for your detailed responses.

---

> > > ### Author Response · Authors · 2026-04-03
> > >
> > > Thank you for your thorough and constructive review. We are grateful that our responses have addressed your concerns. Your insightful questions have significantly strengthened the paper.

---

### Official Review · Reviewer_4NE8 · 2026-03-13

**Soundness:** 3
**Presentation:** 2
**Significance:** 3
**Originality:** 3
**Overall Recommendation:** 4
**Confidence:** 4

**Summary:**

This paper addresses the problems of autoregressive large language models (LLMs) in complex reasoning tasks. The inherent irreversibility of autoregression in the Transformer architecture prevents early errors from being corrected in subsequent steps, ultimately leading to cascading reasoning failure. The authors first decompose LLM reasoning failure into two key stages: first, the potential representational bias generated at the logical anchor point; and second, the explicit cognitive dissonance that erupts at the trigger point after the bias accumulates. Through empirical analysis, the paper quantifies the latency effect between the origin and manifestation of errors.

Based on this core insight, the authors proposed the HEdit (Hindsight Editing) lightweight reasoning enhancement paradigm, which constructs a closed-loop reasoning mechanism of perception-backtracking-editing-regeneration: it identifies key logical anchors of reasoning in real time through untrained modules and detects failure triggers that represent cognitive dissonance; when a reasoning anomaly is detected, it backtracks to the anchor point of the error origin, uses a lightweight trainable editor with less than 0.5% of the parameters to accurately correct the Key-Value (KV) cache corresponding to the anchor point, and finally restarts reasoning based on the corrected state, fundamentally breaking the unidirectional constraint of autoregressive reasoning.

**Compliance With Llm Reviewing Policy:**

Affirmed.

**Final Justification:**

Thank you for the authors for their work and thoughtful reply. After careful consideration, I believe the current score is reasonable.

**Key Questions For Authors:**

My question is the same as the "Weaknesses" section in the previous chapter; please refer to it.

**Limitations:**

Yes, authors discussed the limitations and potential negative societal impact.

**Strengths And Weaknesses:**

1. Strengths

1.1 The method design is supported by solid empirical observations. First, the token distance between the anchor point and the trigger point was quantified through manual auditing, and the two-stage mechanism of inference failure was deconstructed. The core solution is driven by clear phenomena and problems, rather than being designed out of thin air.

1.2 The experimental design is comprehensive and rigorous, covering different model families, parameter scales, strong inference models optimized by RL, and multiple high-difficulty mathematical inference benchmarks. Reasonable baselines such as CoT and naive backtracking were also set, and the sufficient breadth of experiments strongly supports the core conclusions.

1.3 The efficiency and overhead of the method were quantitatively analyzed, including key indicators such as intervention frequency, inference latency, and parameter ratio. This fully verifies the method's design goal of "lightweight and low overhead," rather than sacrificing performance for heap computation.

2. Weaknesses

2.1 The core experiments only cover closed-book mathematical inference tasks, lacking validation on other long-chain inference tasks such as code generation, multi-step logical reasoning, and decision planning. Further analysis of other domains can significantly enhance the generalization ability of the method.

2.2 The ablation experiments only compared the baseline of the random replacement module, lacking comparison with similar alternatives. For example, in the anchor point detection part, could it be compared with rule-based mathematical key information localization or semantic key step recognition schemes?

2.3 Only the effect of single-round correction was verified; it was not verified for scenarios involving multiple anchor point errors or multiple rounds of error triggering in long inference chains. Furthermore, it is unclear whether anchor point correction introduces new problems. Will it destroy the original correct contextual representation? Will it introduce new logical contradictions?

2.4 The format of this paper can be slightly modified to increase rigor. For example, punctuation marks should be added to mathematical formulas by convention; also, figures could be changed to PDF format.

---

> ### Author Rebuttal · Authors · 2026-03-29
>
> We would like to thank you for recognizing the intuitive motivation, comprehensive experiments, and lightweight design of this work. In response to your suggestions for improvement, we have supplemented the relevant experiments and respond to each point as follows:
>
> ---
>
> ### 2.1 New Task Domains
> To verify the generalizability of HEdit, we extended our experiments to **code generation (LiveCodeBench v5)** and **STEM reasoning (GPQA-Diamond)** tasks.
>
> - **Main Results:** As shown in the table below, HEdit significantly improves performance across different architectures (by +5.7% to +7.2%), outperforming CoT and Naive BT (which only improves by 1.8%–2.5%, not shown in table).
>
> **Table R1: Experimental Results on New Tasks (Accuracy %)**
> | Task / Model | Qwen3-14B (Base / +CoT / +Ours) | Llama3.1-8B-Instruct (Base / +CoT / +Ours) | DeepSeek-R1-Distill-Qwen-1.5B (Base / +CoT / +Ours) |
> | :--- | :--- | :--- | :--- |
> | **LiveCodeBench v5** | 46.3 / 48.6(+2.3) / **52.0 (+5.7)** | 11.1 / 12.5(+1.4) / **17.3 (+6.2)** | 13.9 / 16.5(+2.6) / **20.4 (+6.5)** |
> | **GPQA-Diamond** | 58.5 / 61.9(+3.4) / **64.8 (+6.3)** | 33.2 / 35.3(+2.1) / **38.9 (+5.7)** | 34.3 / 36.6(+2.3) / **41.5 (+7.2)** |
>
> - **Zero-shot Transfer:** We directly applied the **KV Editor trained on mathematical datasets** to intervene in code and STEM reasoning tasks (without any fine-tuning). Experiments still show performance improvements of **2.2%–4.5%**. This demonstrates that HEdit captures a **general representational bias** of Transformers when handling complex logic, rather than overfitting to specific tasks.
>
> ---
>
> ### 2.2 Additional Ablation Experiments
> Following your suggestion, we designed an ablation experiment based on linguistic rules: anchors are identified as numbers, variables, and constraint words; triggers are identified as transition words (e.g., *however*), question words (e.g., *why*), and contradiction words (e.g., *inconsistent*).
>
> **Table R2: Rule-based Ablation Experiments (AMC23 Accuracy %)**
> | Model | Ours | +Rule-based Anchor | +Rule-based Trigger |
> | :--- | :--- | :--- | :--- |
> | Qwen3-14B | **75.5** | 70.6 (-4.9) | 70.3 (-5.2) |
> | Llama3.1-8B | **17.0** | 12.7 (-4.3) | 12.2 (-4.8) |
> | DeepSeek-R1-Distill-Qwen-1.5B | **61.8** | 56.7 (-5.1) | 56.1 (-5.7) |
>
> - **Analysis:** Although rule-based methods yield slight improvements of 0.6%–2.2% over the base model, they are significantly inferior to HEdit. Rule-based approaches can only capture explicit key information and fail to cover implicit logical anchors and internal cognitive dissonance within the model. Thus, they only bring marginal gains and perform much worse than the detection method based on the model’s hidden states proposed in this paper.
>
> ---
>
> ### 2.3 Multi-round Correction and Side Effect Analysis of Editing
> - **Multi-round Correction:** This is a highly insightful suggestion. The current HEdit focuses on single-shot high-precision correction. For multi-point errors in long reasoning chains, we plan to explore a **cyclic editing** mechanism in the next stage. We have also discussed this point in Section 5.2 "Limitations and Future Work" of the paper.
> - **Side Effect Analysis:** To verify whether KV editing would damage existing correct representations, we randomly sampled 50 intervention samples and tracked the correctness change before vs. after editing:
>   - **Result Distribution:**
>     - Successfully repaired: 28 samples (56%) — wrong → correct
>     - Unchanged (still wrong): 16 samples (32%) — wrong → wrong
>     - Preserved correctness: 4 samples (8%) — correct → correct
>     - Accidental degradation: 2 samples (4%) — correct → wrong
>   - **In-depth Conclusion:** The vast majority of samples (96%) maintained correctness or were successfully repaired. For the two degraded samples, analysis revealed that the errors were not introduced by editing, but stemmed from new, unrelated computational mistakes during later generation. This proves that HEdit’s local edits do not corrupt existing correct contexts.
>
> ---
>
> ### 2.4 Paper Format
> We fully accept your suggestions. In the camera-ready version, we will:
> - Add punctuation marks to all mathematical formulas.
> - Replace all illustrations with vector PDF format.
> - Optimize the main text layout to ensure rigorous academic presentation.
>
> ---

---

> > ### Author Rebuttal · Reviewer_4NE8 · 2026-04-04
> >
> > Thank you for your work and thoughtful reply. After careful consideration, I believe the current score is reasonable.

---

> > > ### Author Response · Authors · 2026-04-04
> > >
> > > We sincerely appreciate your careful consideration of our rebuttal and your confirmation that the concerns have been resolved. Thank you for your thorough review and insightful feedback.

---

### Decision · Program_Chairs · 2026-04-30

**Decision:**

Accept (regular)

**Comment:**

This paper was well received by the reviewers, who found the work both novel and practically meaningful. First, the paper is grounded in a clear and insightful diagnosis of failure modes in autoregressive reasoning, supported by empirical analysis that distinguishes between error origin (“anchor”) and manifestation (“trigger”), which provides a strong conceptual foundation for the proposed method. Second, the HEdit framework introduces a creative and technically compelling solution by enabling targeted editing of past KV states, effectively breaking the irreversibility of standard autoregressive decoding in a lightweight manner. This idea is both intuitive and innovative, and represents a meaningful step beyond existing test-time reasoning improvements. Third, the empirical results are consistently strong across multiple benchmarks and model scales, with thorough experimentation demonstrating both effectiveness and efficiency, including careful analysis of overhead and ablations validating key design components.

At the same time, reviewers noted a few areas where the work could be further strengthened. In particular, the experimental evaluation would benefit from comparisons against stronger test-time reasoning baselines (e.g., self-consistency or search-based methods) under matched compute budgets, and broader validation beyond mathematical reasoning tasks to demonstrate generality. Additionally, while the proposed anchor/trigger mechanisms are promising, further qualitative and quantitative analysis would help solidify the interpretability claims, and some implementation details and hyperparameter choices could be more clearly justified to improve reproducibility.

Overall, the paper presents a novel, well-motivated, and empirically strong contribution to improving reasoning in LLMs, and the reviewers found the approach both insightful and impactful. Therefore, I recommend acceptance.